# The Usefulness of Scintigraphic Studies in the Assessment of Asymptomatic Bowel Disease in Patients with Primary Antibody Diseases

**DOI:** 10.3390/jcm9040949

**Published:** 2020-03-30

**Authors:** Cinzia Milito, Francesco Cinetto, Valentina Megna, Giuseppe Spadaro, Isabella Quinti, Mauro Liberatore

**Affiliations:** 1Department of Molecular Medicine, Sapienza University of Rome, 00185 Rome, Italy; isabella.quinti@uniroma1.it; 2Department of Medicine-DIMED, University of Padova, 35122 Padova, Italy; francesco.cine@gmail.com; 3Internal Medicine I, Ca’ Foncello Hospital, 31100 Treviso, Italy; 4Department of Radiological, Oncological and Pathological Anatomy Sciences, Sapienza University of Rome, 0161 Rome, Italy; megna_valentina@virgilio.it (V.M.); maurlib@tin.it (M.L.); 5Department of Translational Medical Sciences and Interdepartmental Center for Research in Basic and Clinical Immunology Sciences, University of Naples Federico II, 80131 Naples, Italy; spadaro@unina.it

**Keywords:** primary immunodeficiency diseases, inflammatory bowel disease, scintigraphy with human polyclonal immunoglobulin G labeled, scintigraphy with white blood cells labeled by 111 Indium-oxinate, IgG trough levels

## Abstract

Enteropathy may be the first presentation of immunodeficiency or it may occur during the course of the disease and in association with malabsorption in patients affected by primary antibody diseases. For these patients, immunoglobulin G (IgG) replacement therapy prevents infectious and non-infectious complications. Nonetheless some patients cannot achieve optimal IgG trough levels, even when treated with high Ig doses in absence of protein-losing syndromes. We investigated seven patients affected by common variable immunodeficiencies (CVIDs) and treated with high Ig doses (600–800 mg/kg/month) showing low IgG trough level. Patients underwent abdominal scintigraphy with human polyclonal immunoglobulin G labeled with 99mTc and with white blood cells labeled by 111 Indium-oxinate to investigate asymptomatic bowel inflammation. A concentration of labeled leukocytes in abdominal segments greater than that observed with human polyclonal immunoglobulin G was evident only in one patient. In five patients a slight concentration of both radiopharmaceuticals was reported, due to mild intestinal inflammatory response. These data might be related to mild increase of capillary permeability in the absence of inflammation leukocyte mediated. This study discloses a new cause of IgG-accelerated catabolism due to inflammatory bowel conditions without diarrhea in CVID patients.

## 1. Introduction

Common variable immunodeficiencies (CVIDs) are the most common symptomatic primary immunodeficiencies in adults. The disease is characterized by low levels of serum immunoglobulin G (IgG), IgA and/or IgM, impaired specific antibody production in response to vaccination, and higher increased risk of bacterial infections associated to non-infectious complications. Moreover, a diagnosis of CVID can be performed only after excluding secondary causes of hypogammaglobulinemia (infection, protein loss, medication, malignancy) [1].

Commonly, CVID patients present respiratory tract infections, autoimmunity, cancers, and gastrointestinal (GI) diseases.

The frequency of gastrointestinal manifestations of CVIDs is variable (20–60%) [2]. GI diseases include chronic atrophic gastritis, pernicious anemia, nodular lymphoid hyperplasia, chronic diarrhea, inflammatory bowel disease, celiac-like disease, lymphomas, gastric adenocarcinoma, and non-specific malabsorption [3]. Gastrointestinal inflammatory diseases can be present even in pauci-symptomatic patients [4]. Enteropathy may be the first clinical presentation of immunodeficiency [5] or it may occur during the course of the disease affecting large and/or small bowels and it might be associated with malabsorption [6].

Immunoglobulin replacement therapy is the standard treatment for patients with CVIDs in order to avoid infectious and non-infectious complications [7]. The majority of CVID patients are on either intravenous (IVIG) or subcutaneous (SCIG) treatment [8,9,10].

Although there is a wide variation in treatment practice, international guidelines on immunoglobulin replacement suggest a cumulative monthly dose of 400 mg/Kg to achieve IgG trough levels of at least 5 g/L. Several studies [11] suggest achieving higher levels of around 6 to 8 g/L to control infections.

Each patient requires an individual dose of therapeutic immunoglobulin to prevent breakthrough infections. Consequently, IgG trough levels vary between patients [12,13]. Despite these rules and the attempt towards an individualized therapy, some patients cannot achieve optimal IgG trough levels, even when high Ig doses (600–800 mg/kg/month) are administered in absence of evident causes of protein-losing syndromes.

In this work, we studied a group of patients with primary antibody deficiencies and low IgG trough levels despite high IgG replacement doses. The aim was to investigate the possibility that the poor efficacy of Ig replacement might be caused by an asymptomatic bowel inflammation in the absence of enteropathy-related symptoms (chronic diarrhea and malabsorption) using abdominal scintigraphy with human polyclonal immunoglobulin G labeled with 99mTc (99mTc-HIG) and with white blood cells labeled by 111 Indium-oxinate (white blood cells (WBCs)) and gastric endoscopy to exclude malabsorption.

## 2. Materials and methods

### 2.1. Patients

We studied seven patients (6 males and 1 female) affected by CVIDs diagnosed according to current European Society for Immunodeficiencies (ESID) criteria [1]. In all patients underlying monogenic molecular defects were excluded. All patients were regularly followed in inpatient and daycare settings by university hospitals working as referral centers for adult primary immune deficiencies in Rome and included in the Italian Network for Primary Immunodeficiency IPINET Italian Registry for CVID.

During a period of 1 year, we observed low IgG trough levels (380 ± 100 mg/dL) in these patients on intravenous IgG replacement (IVIG) therapy at 400 mg/kg/month. For this reason, according to the international guidelines, we decided to increase the intravenous IgG replacement therapy at a cumulative dose of 600 mg/kg/month receiving IVIG infusion every 2 weeks.

Despite the increase of the cumulative dosage, after 6 months, IgG trough levels were <500 mg/dL (343 ± 54 mg/dL) in six patients. An IgG trough level >700 mg/dL was only reached by one patient, whom we decided to analyze as the control (patient 7).

For each patient evident causes of protein-losing were excluded (chronic diarrhea, nephropathy, use of drugs, lymphoproliferative syndromes) and serum protein and albumin levels were within the range of normality. According to the current ESID criteria, patients included in the study did not show profound T-cell deficiency. The mean count of CD4+ T cells was 457 (absolute numbers/microliter) (SD ±214), whereas the mean percentage of naïve CD4+ T cells was 28% (SD ±12). No patient included in our study had lymphedema and/or intestinal lymphangiectasia. Immunological, laboratoristic, and clinical characteristics as well as clinical phenotype are summarized in Table 1 according to Chapel et al.’s [14] proposal of CVID patients.

In order to investigate the presence of asymptomatic enteropathy, all patients underwent abdominal scintigraphy with human polyclonal IgG labeled with 99mTc (99mTc-HIG, Mallinckrodt, Petten, The Netherlands) and with white blood cells (WBCs) labeled by 111 Indium-oxinate. In addition, all patients underwent gastric endoscopy in order to exclude malabsorption, cancer, celiac, or celiac-like disease.

All participants provided written informed consent and the study was approved by the institutional review board at Sapienza University of Rome.

### 2.2. Scintigraphic Studies

#### 2.2.1. White Blood Cell Scintigraphy

All patients underwent a white blood cell (WBC) scintigraphic scan of the abdomen. The separation and labeling technique by 111 Indium-oxinate used for the white blood cells was done as previously described by Roca et al. [15]. Briefly, 9 mL of acid-citrate-dextrose were drawn into a 60 mL plastic syringe and mixed with 51 mL of the patient’s blood. Then, 15 mL of the content of the syringe were centrifuged at 2000× *g* for 10 min to obtain cell-free plasma (CFP). After, 4.5 mL of 2-hydroxyethyl starch (10% HES, pharmaceutical grade) were added to the 60 mL syringe to allow erythrocyte sedimentation for 45 min. Then, the supernatant of the syringe was centrifuged at 150× *g* for 5 min. The mixed leukocyte pellet was re-suspended in 1 mL of saline and labeled with 20 MBq of 111 Indium-oxinate (^111^In-oxin, Mallinckrodt, Petten, The Netherlands) by incubation for 10 min at room temperature. After the incubation 10 mL of saline were added, and the cells were centrifuged at 150× *g* for 5 min. After centrifugation, the supernatant containing unbound ^111^In-oxine was removed and the pellet was re-suspended in 3–5 mL of CFP. The labeled leukocyte pellet was re-suspended in cell-free plasma and re-injected into the patients. A sample of these cells was submitted to Trypan blue test to assess the viability of the labeled cells. A static acquisition of the abdomen was performed in all the patients 20–24 h after the injection of the labeled leukocytes in anterior and posterior view by means of a computerized gamma camera (Xeleris, G.E. Healthcare, Milan, Italy) equipped with low-energy all-purpose collimators.

#### 2.2.2. Scintigraphy with Human Polyclonal Immunoglobulin G Labeled with 99mTc

Seven days after the first scan, patients were submitted to abdominal scintigraphy with human polyclonal immunoglobulin G labeled with 99mTc (99mTc-HIG). Static scintigraphic images of the abdomen were acquired 4 h after the injection of 370 MBq of the radiopharmaceutical in anterior and posterior views. The scintigraphic images were reviewed by a specialist who was unaware of the patient’s clinical status.

#### 2.2.3. Evaluation of Bowel Abnormalities on Scintigraphy with White Blood Cells and With Human Polyclonal IgG

On the basis of intestinal leukocyte and human polyclonal immunoglobulin G uptake on 4 regions (small bowel, ascending, transverse, and descending colon) the scans were classified by a 4-point scale, for each region, as follows: O = no uptake; 1 = faint uptake; 2 = intermediate uptake; 3 = strong uptake. Only the scans scoring 1, 2 or 3 were accepted as really being positive. For each patient intestinal WBC and 99mTc-HIG uptake were evaluated by a total score, defined as the addition of the uptake scores assigned to small bowel, ascending, transverse, and descending colon for each radiopharmaceutical.

### 2.3. Statistical Analysis 

CD4+ T cell counts were summarized as means with standard deviation (SD). The human polyclonal immunoglobulin G total score and the IgG trough levels were plotted and the level of correlation between these two variables was explored using the Spearman correlation analysis, considering *p*-values below 0.05 as statistically significant.

## 3. Results

### Comparison Between Scintigraphy with WBCs and 99mTc-HIG

A comparison of results obtained by scintigraphy with WBCs and 99mTc-HIG is shown in Table 2.

Both techniques provided positive results in six patients with IgG trough levels lower than <500 mg/dL while the control patient had no segments involved. We analyzed a total of 24 segments from six patients.

In 9/24 bowel segments there was no uptake of both radiopharmaceuticals. In 7/24 segments we observed the same positive uptake of WBCs and 99mTc-HIG; in 2/24 segments the leukocytes uptake exceeded the uptake of immunoglobulins.

In 1/4 segments the concentration of 99mTc-HIG was greater than that obtained with WBCs and in 5/24 segments we observed only a 99mTc-HIG uptake.

The ascending and descending colon were more frequently involved in the uptake of 99mTc-HIG compared to small bowel and transverse colon (six patients and five patients, respectively vs. one patient and two patients, respectively).

Evaluating ascending and descending colon by 99mTc-HIG uptake we found that in the first segment 4/6 patients had a score 1 and 2/6 patient had a score 2; in the second segment, 3/6 patients had a score 1 and 3/6 patients had a score 2.

An uptake of both radiopharmaceuticals in the small bowel was present only in patient 1.

The involvement of transverse colon was recorded in two patients. Patient 2 had an uptake of 99mTc-HIG (score 1) while patient 4 had an uptake of both radiopharmaceuticals with a score 1.

WBCs and 99mTc-HIG scores are summarized in Figure 1.

The analysis of correlation between human polyclonal immunoglobulin G total score and IgG trough levels showed a non-significant trend for an inverse correlation between the two variables (Spearman’s correlation coefficient: −0.600, *p* = 0.154) (Figure 2).

Patient 1, affected by autoimmune thyroid disease, had an involvement of the small bowel. In particular, the small bowel showed a 3-point uptake of radiolabeled WBCs and a 1-point uptake of 99mTc-HIG. In the ascending colon radiolabeled WBCs had a 3-point uptake and 99mTc-HIG had a 1-point uptake. In the transverse colon there was no uptake of both radiopharmaceuticals. The descending colon showed a 1-point uptake of both radiopharmaceuticals. The endoscopy shows chronic gastritis *Helicobacter pylori* positive with gastric intestinal metaplasia, hiatal hernia, and erosive esophagitis.

Patient 2 showed no uptake of both radiopharmaceuticals in the small bowel. In the ascending colon both radiopharmaceuticals had a 2-point uptake. The transverse colon had a 0-point uptake of WBCs and a 1-point uptake of 99mTc-HIG. In the descending colon we observed no uptake of WBCs and a 2-point uptake of 99mTc-HIG. The endoscopy showed chronic gastritis.

Patient 3, with an autoimmune thyroid disease, showed no uptake of both radiopharmaceuticals in the small bowel. In the ascending colon both radiopharmaceuticals had a 1-point uptake. In the transverse colon there was no uptake of both radiopharmaceuticals. In the descending colon there was no uptake of WBCs while there was a 1-point uptake of 99mTc-HIG (Figure 3). The endoscopy showed chronic gastritis.

Patient 4 showed no uptake of both radiopharmaceuticals in the small bowel. In the ascending and transverse colon both radiopharmaceuticals showed a 1-point uptake. In the descending colon we observed a 1-point uptake of WBCs and a 2-point uptake of 99mTc-HIG. The endoscopy did not show any alteration.

Patient 5 showed no uptake of both radiopharmaceuticals in the small bowel. In the ascending colon both radiopharmaceuticals showed a 1-point uptake, but no uptake of both radiopharmaceuticals in the transverse colon. In the descending colon both radiopharmaceuticals had a 1-point uptake. The endoscopy showed chronic gastritis.

Patient 6 showed no uptake of both radiopharmaceuticals in the small bowel. In the ascending colon there was no uptake of WBCs and a 2-point uptake of 99mTc-HIG. We did not observe uptake of both radiopharmaceuticals in the transverse colon. In the descending colon there was no uptake of WBCs and a 2-point uptake of 99mTc-HIG (Figure 4). The endoscopy showed chronic gastritis.

Patient 7 (control) showed no uptake of both radiopharmaceuticals in the four segments studied (Figure 5). The endoscopy showed hiatal hernia.

## 4. Discussion

It is well known that CVID patients show a broad range of clinical manifestations. In particular, a recent study proposed to classify CVID patients into five distinct clinical phenotypes (infections only, cytopenias, polyclonal lymphoproliferation, enteropathy, and lymphoid malignancy) [14] and to adjust the doses of immunoglobulin replacement therapy (IgRT) required to keep a patient free of infections, according to the various clinical phenotypes. It has been reported that patients with enteropathy, cytopenia, and polyclonal lymphoproliferation should be treated with significantly higher doses of immunoglobulin to prevent infections [5]. Other studies underlined that each patient requires an individual immunoglobulin dose to maintain a target IgG level higher than the level proposed by guidelines (6 to 8 g/L) [12,13,16] and that this individual immunoglobulin dose should be adjusted in line with the clinical phenotype. Patients with chronic lung disease and bronchiectasis or inflammatory bowel disease often require higher doses of IgG and may not reach the desired trough level. Therefore, the doses of immunoglobulin required to keep a particular patient free of bacterial infections are specific for that patient.

In those patients that cannot achieve optimal IgG trough levels, an intestinal protein-losing syndrome should be excluded. As an asymptomatic bowel inflammation might be the cause of a protein-loss, our study was designed to diagnose this condition in CVID patients on immunoglobulin replacement with sub-optimal serum IgG levels and without obvious symptoms of enteropathy. Imaging of bowel inflammation can be performed with several methods, even if nuclear medicine techniques produce the best results in terms of diagnostic accuracy [17]. As far as we are aware, the only way to distinguish between a leukocyte infiltration of the intestine and the intestinal loss of immunoglobulins is by using this scintigraphic investigation technique. Indeed, this distinction cannot be obtained with magnetic resonance imaging (MRI). A histologic approach to the whole bowel might also be more invasive and difficult, and a good correlation between scintigraphic and histologic findings has already been suggested in different contexts [18], although there are no comparable data on scintigraphy and histology in the specific setting of a CVID patient. Furthermore, it is worth pointing out that diagnostic radiation doses expose patients to stochastic and not deterministic risks, largely compensated by the benefits provided to the patients themselves when an asymptomatic protein-losing enteropathy is disclosed.

White blood cell scan as well as 99mTc-labeled non-specific polyclonal immunoglobulin G have been widely employed in the study of inflammatory bowel diseases with satisfactory results. It has been also shown that labeled white blood cells are more sensitive than labeled polyclonal IgG in detecting bowel inflammation [19].

In our study, a concentration of labeled leukocytes in abdominal segments greater than that observed with HIG was evident only in one of our seven patients. In the other patients we found a slight concentration of both of the radiopharmaceuticals in seven segments of five patients, probably due to the presence of mild intestinal inflammatory response. The main segments involved in the uptake of both radiopharmaceuticals were ascending and descending colon. However, there were six segments in four patients in which a concentration of HIG did not correspond to a leukocyte’s concentration of equal intensity.

This latter finding could be related to a modest increase in capillary permeability that only permits protein extravasation, without a significant leukocytes’ concentration in the site. Our study suggests that a condition of protein dispersion without the presence of a leucokyte-mediated inflammation could exist. In fact, it is well known that the uptake mechanism of HIG is non-specific and it relies on pathophysiological changes occurring at the inflammation site such as the enhanced vascular permeability, blood flow, and transudation of plasma proteins in absence of inflammation leukocyte-mediated [20]. On the other hand, WBCs have been widely used in the study of Crohn’s disease both for the evaluation of the extent and inflammatory activity of the disease, and for the study of abscesses and abdominal fistulas that can complicate the disease [21]. In fact, WBC uptake depends on the enhanced influx of granulocytes to the infection/inflammation site mediated via immunological binding to cellular antigens and chemotactic peptides (f-Met-Leu-Phe) [22,23], cytokines (interleukin 1 (IL-1), IL-8, Platelet factor 4 PF-4) [24,25,26], and complement factors (C5a, C5adR) [27].

Some patients included in the study presented CVID-related complications as Chronic lung disease CLD and bronchiectasis. It might be speculated that, in presence of bronchiectasis, lungs might be the site where IgG are mainly concentrated, with consequently low IgG trough levels. In the present study, we did not focus our imaging on the lungs, but we might hypothesize that inflammatory or infectious lung involvement should be related to a higher uptake of leukocytes than to immunoglobulin utilization. Moreover, IgG recruitment in the lungs would likely be related to active infection rather than to subclinical inflammatory or non-inflammatory conditions as barrier permeability leading to IgG loss, as suggested by patient 7 in our study.

Furthermore, on the basis of the results obtained by bowel scintigraphy, our patients were shifted from IVIG to SCIG treatment with a gradual and effective increase of IgG trough levels. It is known that the kinetics of IgG distribution depend on the route of administration. Indeed, when IgG are administered subcutaneously, they are slowly absorbed and redistributed during concentration-dependent catabolism [28]. Thanks to this mechanism, IgG may reduce bowel inflammation associated to an inadequate barrier permeability. For these reasons, in CVID patients with bowel involvement, replacement therapy with subcutaneous IgG may be considered as the treatment of choice [28].

Finally, despite not being statistically significant, the analysis of correlation between human polyclonal immunoglobulin G total score and IgG trough levels in our patients showed a trend for an inverse correlation between the two variables. However, in this pilot study the statistical power to detect a significant correlation was limited. Thus, the inverse relationship between human polyclonal immunoglobulin G total score at scintigraphy and IgG trough levels should be confirmed in specifically designed studies based on a larger number of cases.

## 5. Conclusions

This study discloses a new cause, previously hypothesized but not yet demonstrated, of accelerated IgG catabolism due to an inflammatory bowel condition without diarrhea in patients with antibody deficiencies and low IgG trough levels. In all patients, except one, we found a high concordance between ^111^In-oxine labeled leukocyte scan and 99mTc-labeled human immunoglobulin G uptake as a consequence of light bowel inflammation. In four patients, six intestinal segments showed a different concentration between labeled leukocyte and human immunoglobulin G, suggesting a possible mild increase of capillary permeability in absence of leucokyte-mediated inflammation. Thus, our study suggests that an intestinal loss of immunoglobulins should be hypothesized in patients not achieving adequate IgG trough levels despite appropriate intravenous replacement therapy, even in absence of a symptomatic enteropathy. The evidence of a subclinical intestinal loss of IgG would then have clear therapeutic implications, driving the shift from intravenous to subcutaneous Ig replacement therapy.

## Figures and Tables

**Figure 1 jcm-09-00949-f001:**
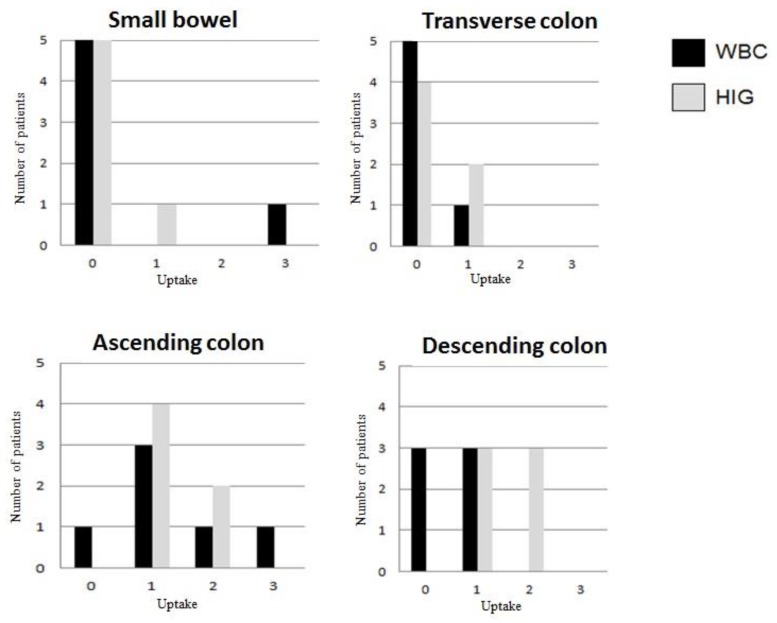
Comparison of white blood cells (WBC) and 99mTc-HIG scores in the small bowel, ascending, transverse, and descending colon. X axis: uptake of WBCs (black bars) and 99mTc-HIG (grey bars) by a 4-point scale (0 = no uptake; 1 = faint uptake; 2 = intermediate uptake; 3 = strong uptake). Y axis: number of patients. Patient 7 is not included.

**Figure 2 jcm-09-00949-f002:**
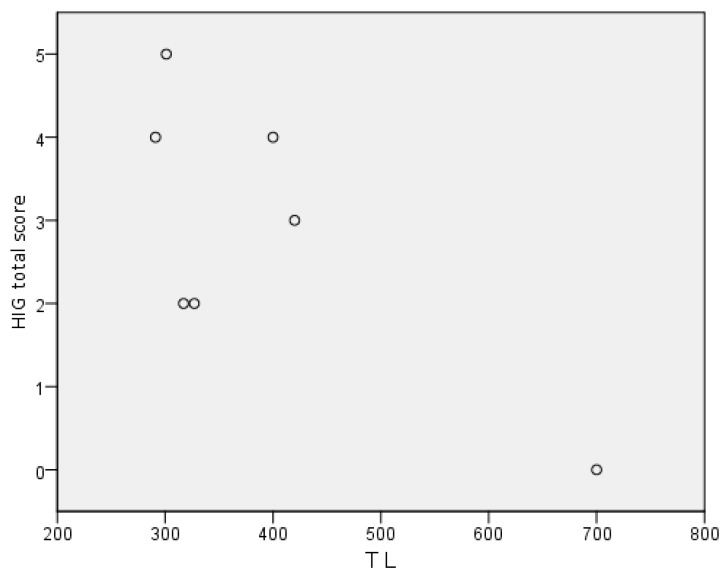
Scatter plot of human polyclonal immunoglobulin G total score (HIG total score) and IgG trough level (TL). Spearman’s correlation coefficient: −0.600, *p* = 0.154.

**Figure 3 jcm-09-00949-f003:**
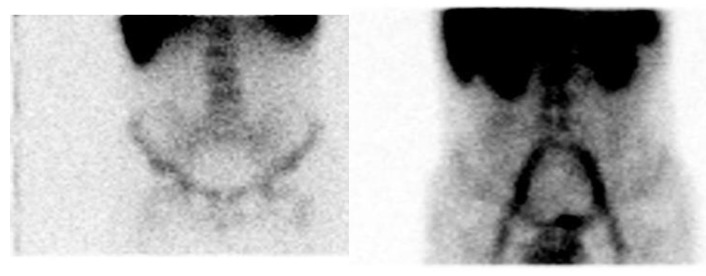
Comparison of ^111^In-oxine labeled leukocyte scan (left) and 99mTc-labeled human immunoglobulin G scan (right) in the same patient (anterior view). Pathological uptake of both radiopharmaceuticals on the ascending colon; only labeled immunoglobulin scan shows pathological uptake on descending colon.

**Figure 4 jcm-09-00949-f004:**
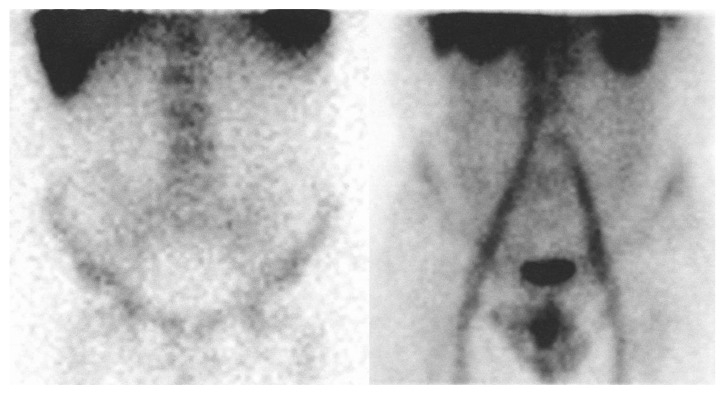
Comparison of ^111^In-oxine labeled leukocyte scan (left) and 99mTc-labeled human immunoglobulin G scan (right) in the same patient (anterior view). The labeled immunoglobulin scan shows pathological uptake of the radiopharmaceutical on the ascending and descending colon, while leukocyte scan does not show areas of pathological leukocyte concentration.

**Figure 5 jcm-09-00949-f005:**
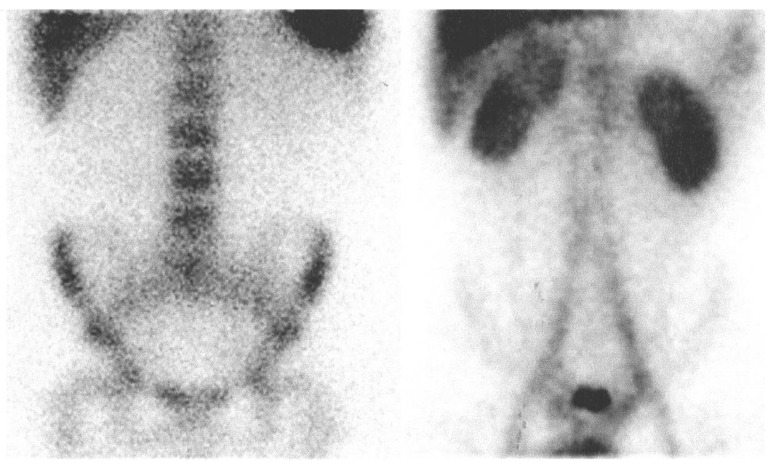
Comparison of ^111^In-oxine labeled leukocyte scan (left) and 99mTc-labeled human immunoglobulin G scan (right) in the same patient (anterior view). The images do not show areas of pathological concentration of both radiopharmaceuticals.

**Table 1 jcm-09-00949-t001:** (**a**) Immunological and (**b**) clinical data of seven common variable immunodeficiency (CVID) patients.

**(a)**										
**Pts**	**Age** **(yrs)**	**Gender**	**Onset Age** **(yrs)**	**IgG TL** **(mg/dL)**	**IgA** **(mg/dL)**	**IgM** **(mg/dL)**	**Weight** **(kg)**	**Intervals Therapy** **(days)**	**Serum Protein** **(mg/dL)**	**Albumin Level** **(mg/dL)**
**1**	74	F	39	420	<6	42	55	15	6.7	4.6
**2**	45	M	31	301	<6	<4	80	21	6.6	4.6
**3**	50	M	40	317	<6	<4	55	10	6.2	4.3
**4**	37	M	20	400	<6	<4	80	21	6.5	3.9
**5**	74	M	35	327	<6	<4	80	21	6.4	4.5
**6**	54	M	28	291	78	71	62	15	6.4	4.6
**7**	44	M	38	700	<6	17	78	21	7.3	4.3
**(b)**										
**Pts**		Chapel et al.’s Phenotype	Chronic Lung Disease		Splenomegaly	Autoimmunity		Bronchiectasis	Chronic Lymphatic Hyperplasia	Sinusitis
**1**		Polyclonal lymphoproliferation	Yes		No	Yes (thyroid disease)		Yes	Yes	Yes
**2**		Polyclonal lymphoproliferation	Yes		Yes	Yes (vitiligo)		Yes	Yes	Yes
**3**		Cytopenia	No		No	Yes (thyroid disease)		No	Yes	Yes
**4**		Infection only	No		No	No		Yes	No	No
**5**		Polyclonal lymphoproliferation	No		Yes	No		No	Yes	No
**6**		Cytopenia	No		Yes	Yes (thyroid disease)		No	No	No
**7**		Infection only	No		No	No		Yes	No	Yes

IgG TL: Immunoglobulin G trough level; yrs: years.

**Table 2 jcm-09-00949-t002:** Comparison of uptake of white blood cells (WBCs) and human polyclonal immunoglobulin G labeled with 99mTc (99mTc-HIG) in the small bowel and in ascending, transverse, and descending colon in seven patients studied by scintigraphy. Evaluation of bowel abnormalities by a 4-point scale, for each region, as follows: O = no uptake; 1 = faint uptake; 2 = intermediate uptake; 3 = strong uptake. Only the scans scoring 1, 2 or 3 were accepted as really being positive.

	Small Bowel		Ascending Colon		Transverse Colon		Descending Colon	
Patient	WBC	HIG	WBC	HIG	WBC	HIG	WBC	HIG
**1**	3	1	3	1	0	0	1	1
**2**	0	0	2	2	0	1	0	2
**3**	0	0	1	1	0	0	0	1
**4**	0	0	1	1	1	1	1	2
**5**	0	0	1	1	0	0	1	1
**6**	0	0	0	2	0	0	0	2
**7**	0	0	0	0	0	0	0	0

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
