# Peer review of "The Usefulness of Scintigraphic Studies in the Assessment of Asymptomatic Bowel Disease in Patients with Primary Antibody Diseases"

_jcm, 2020, doi:10.3390/jcm9040949_

Round 1

Reviewer 1 Report

Major points

The authors claim they have identified IgG-hypercatabolism due to inflammatory bowel disease without diarrhea in 5 patients, whose IgG-levels remained low despite immunoglobulin replacement.

They have to exclude that those patients had intestinal lymphangiectasia or another protein-loosing enteropathy. Were serum and fecal concentration of alpha(1)-antitrypsin tested? Are serum protein and albumin levels available? Did any of the patients had lymphedema?

Peripheral lymphocyte and lymphocyte subset (at least CD4+ T cells and if possible naïve CD4+T cells) counts should be reported. This would show that the authors comply with current ESID criteria for the diagnosis of CVID. Further, normal counts would exclude intestinal lymphangiectasia, which I suspect in at least some of the patients.

Minor points

Introduction can be improved by stating that  for CVID diagnosis secondary hapogammaglobulinemia needs to be excluded

Have the authors exluded monogenic PID and esp. b2 microglobulin deficiency?

Author Response

The authors claim they have identified IgG-hypercatabolism due to inflammatory bowel disease without diarrhea in 5 patients, whose IgG-levels remained low despite immunoglobulin replacement.

We thank Rev1 for her/his suggestions/comments, that helped us in improving the quality of the manuscript.

They have to exclude that those patients had intestinal lymphangiectasia or another protein-loosing enteropathy. Were serum and fecal concentration of alpha (1)-antitrypsin tested? Are serum protein and albumin levels available? Did any of the patients had lymphedema?

Answer: In M&M we underlined that “For each patient evident causes of protein-losing were excluded (chronic diarrhea, nephropathy, use of drugs, lymphoproliferative syndromes)”. Patients underwent gastroschopy to exclude celiac disease or celiac-like disease. Serum and fecal concentration of alpha (1)-antitrypsin was not tested. For each patient serum protein and albumin levels are available. These levels are within the range of normality (serum protein: 6.0 – 8.2 gr/dL; albumin level: 3.5 - 5.5 gr/dL) and have been included in Table 1 section a. No patient included in the study had a condition of lymphedema.

Peripheral lymphocyte and lymphocyte subset (at least CD4+ T cells and if possible naïve CD4+T cells) counts should be reported. This would show that the authors comply with current ESID criteria for the diagnosis of CVID. Further, normal counts would exclude intestinal lymphangiectasia, which I suspect in at least some of the patients.

Answer: Peripheral lymphocyte and lymphocyte subset are available for each patient. For each patient the diagnosis of CVID was done according to current ESID criteria. No patient included in the present study showed profound T-cell deficiency. The mean count of CD4+ T cells was 457 numbers/microliter (SD ± 214) numbers/microliter, while the mean count of naïve CD4+T cells was 28% (SD ± 12). No patient included in the study had an intestinal lymphangiectasia.

Reviewer 2 Report

The basic concept and idea behind this study is interesting. It is also important clinically. However there are some questions that remained unanswered. The following issues and questions needs to be addressed.

  1. Authors state that one of the patients with relatively good IgG level serves as a control. It is probably not enough. If a proper control is desirable these should be about the same number of patients with adequate IgG through levels.
  2. Obviously the use of radioactive substances is not a preferable mode of diagnosis in patients susceptible to cancer and proliferative diseases. Is it possible to achieve a diagnosis of enteritis by MRE?
  3. In some of the patients pulmonary involvement in the form of CLD and bronchiectasis is describe. Is it possible that this is the site where Ig's are "consumed", utilized? There is no mention to this option in the presented study.
  4. Most of the reported gut involvement is described as mild or mild-moderate, can we really account that as the cause for such a persistently low Ig's level in spite of both increase in dose and shortened intervals between infusions?

Author Response

Rewier 2: Minor points

We thank Rev2 for her/his suggestions/comments, that helped us in improving the quality of the manuscript.

Q: Introduction can be improved by stating that for CVID diagnosis secondary hapogammaglobulinemia needs to be excluded.

A: The introduction was modified adding this suggestion and Reference 1 was modified.

Q: Have the authors exluded monogenic PID and esp. b2 microglobulin deficiency?

A: In all patients underlying monogenic molecular defect have been excluded.

The basic concept and idea behind this study is interesting. It is also important clinically. However there are some questions that remained unanswered. The following issues and questions needs to be addressed.

  1. Authors state that one of the patients with relatively good IgG level serves as a control. It is probably not enough. If a proper control is desirable these should be about the same number of patients with adequate IgG through levels. Thank you for your observation. Usually, in a clinical trial you have to enroll the same number of patients and controls. In this case, it was not possible to include the same number of patients because there is no such low radiation dose to rule out any risk. Moreover, by law, administration of radioactive substances and radiation exposure is unjustified for asymptomatic
  2. Obviously the use of radioactive substances is not a preferable mode of diagnosis in patients susceptible to cancer and proliferative diseases. Is it possible to achieve a diagnosis of enteritis by MRE? Thank you for your suggestion. We included these sentences in Discussion. “There is no such low radiation dose to rule out any risk. For this reason the use of diagnostic imaging based on the administration of radioactive substances is allowed, by law, only when the resulting information is not obtainable with other methodologies.  The aim of our study was not to diagnose enteritis, but to distinguish between a leukocyte infiltration of the intestine and the intestinal loss of immunoglobulins. This distinction cannot be obtained with magnetic resonance imaging. Thus, the information necessary to our research are not achievable with investigations other than those employed. Furthermore, it is worth pointing out that diagnostic radiation doses expose patients to stochastic and not deterministic risks, largely compensated by the benefits provided to the patients themselves”.
  3. In some of the patients pulmonary involvement in the form of CLD and bronchiectasis is describe. Is it possible that this is the site where Ig's are "consumed", utilized? There is no mention to this option in the presented study. Thank you for your observation. We added this sentence in the manuscript in Discussion. We have no data on lung involvement because our study was focused on abdomen. We don’t believe that immunoglobulins are “consumed” at lung sites; the lung involvement characterized in CVID patients by CLD and bronchiectasis should be related to a higher uptake of leucocytes than to immunoglobulin’s utilization.
  4. Most of the reported gut involvement is described as mild or mild-moderate, can we really account that as the cause for such a persistently low Ig's level in spite of both increase in dose and shortened intervals between infusions? According to us, after excluding all other causes of protein-losing, the persistently low Ig's level, is related to inflammatory bowel disease.

Reviewer 3 Report

In the study the authors describe 7 patients with CVID who have insufficient trough IgG levels despite adequate ivIG replacement therapy and underwent scintigraphy. As the authors state correctly, insufficient IgG replacement worsens the long term outcome of CVID patients. Therefore, in principle, this topic is of high interest. However, the study has major limitations:

- The authors state that „For each patient evident causes of protein-losing were excluded“. However important information such as the calprotectin levels in the stool are missing. It would be important to see how such widely available tests correlate to more complex tests as scintigraphy.
- The authors speculate that „asymptomatic bowel inflammation might be the cause of a protein-loss“. This idea is supported by the scintigraphy findings. However, the clinical and therapeutical implications remain unclear. The authors should include 1) when to consider scintigraphy and 2) how to improve the patients‘ therapy based on the results. The clinical implications of this study remain unclear.

Further there are some minor comments:
- In this small subset of CVID patients, there seems to be a male predominance. Further information about the patients‘s phenotype and the genetics would therefor be helpful to interpret these results.
- In Figure 1, the labeling of the axis is missing.
- In Figure 2, the authors used a Pearson correlation. For a Pearson correlation, each variable should be continuous. The HIG total score (sum of four locations with O = no uptake; 1 = faint uptake; 2) intermediate uptake; 3) strong uptake for each location) is not continuous. Therefore Pearson correlation cannot be used. Since this is an essential figure of the manuscript, no conclusion should be drawn prior to revision of the statistics.

Author Response

Reviewer 3:

We thank Rev3 for her/his suggestions/comments, that helped us in improving the quality of the manuscript.

In the study the authors describe 7 patients with CVID who have insufficient trough IgG levels despite adequate IVIG replacement therapy and underwent scintigraphy. As the authors state correctly, insufficient IgG replacement worsens the long term outcome of CVID patients. Therefore, in principle, this topic is of high interest. However, the study has major limitations:

Q: The authors state that „For each patient evident causes of protein-losing were excluded“. However important information such as the calprotectin levels in the stool are missing. It would be important to see how such widely available tests correlate to more complex tests as scintigraphy. 

A: Calprotectin levels in the stool were available only for 4 patients and were negative.

- The authors speculate that „asymptomatic bowel inflammation might be the cause of a protein-loss“. This idea is supported by the scintigraphy findings. However, the clinical and therapeutical implications remain unclear. The authors should include 1) when to consider scintigraphy and 2) how to improve the patients‘ therapy based on the results. The clinical implications of this study remain unclear. Thank you for your observation.

1)The aim of our study was not to diagnose enteritis, but to distinguish between a leukocyte infiltration of the intestine and the intestinal loss of immunoglobulins. This distinction cannot be obtained with magnetic resonance imaging. Thus, the information necessary to our research are not achievable with investigations other than those employed. We added this sentence in Discussion.

2) On the basis of the results obtained by scintigraphy our patients shifted from IVIG to SCIG treatment.  We know that the kinetics of IgG distribution depend on the route of administration. In fact, when administered subcutaneously, IgG are slowly absorbed and redistributed during concentration-dependent catabolism. Thank to this mechanism, IgG reduces bowel inflammation associated to an inadequate barrier permeability. Moreover, we suggest that for patients with CVID and bowel involvement, replacement therapy with subcutaneous IgG may be the treatment of choice. Lastly, these patients might be considered candidates for therapy with subcutaneously IgG therapy. Previously, we did not included these results because of not related to our findings but now we have included in Conclusions.

Further there are some minor comments:
Q: In this small subset of CVID patients, there seems to be a male predominance. Further information about the patients‘s phenotype and the genetics would therefore be helpful to interpret these results.

A: Patients underwent genetic tests to identify monogenic defects that were negative. The clinical phenotype for each patient was collected according to Chapel’s et al. proposal (Chapel H, Lucas M, Patel S, Lee M, Cunningham-Rundles C, Resnick E, et al. Confirmation and improvement of criteria for clinical phenotyping in common variable immunodeficiency disorders in replicate cohorts. J Allergy Clin Immunol Pract. (2012) 130:1197–8. doi: 10.1016/j.jaci.2012.05.046) and have been added in Table 1 section b.

- In Figure 1, the labeling of the axis is missing.

Thank you for your suggestion. We added the labeling of axis in Figure 1.

- In Figure 2, the authors used a Pearson correlation. For a Pearson correlation, each variable should be continuous. The HIG total score (sum of four locations with O = no uptake; 1 = faint uptake; 2) intermediate uptake; 3) strong uptake for each location) is not continuous. Therefore, Pearson correlation cannot be used. Since this is an essential figure of the manuscript, no conclusion should be drawn prior to revision of the statistics.

We agree and have revised the statistics using a Spearman correlation analysis which is more appropriate when categorical ordinal variables are used. This is also now clarified in the methods section. The revised results are reported in the text and in the legend to Figure 2. The correlation analysis showed a significance level of 0.154. The number of cases was however very low and therefore the statistical power markedly limited. Overall, the data suggest a non-significant trend that should be confirmed in larger studies. We have acknowledged this as a study limitation in the discussion section.

Round 2

Reviewer 1 Report

Diagnosis of protein-loosing enteropathy is extremely unlikely given the provided serum protein-levels. Further, the authors now state the made the diagnosis of CVID considering the current criteria.

In my opinion this work suggests a previously unidentified diagnostic value of scintigraphic studies in CVID. Quiescent IBD could be identified through presented scintigraphic studies and represent a cause of poor increase of IgG despite antibody replacement.

This could be clinically extremely relevant and the findings are novel. I therefore think the manuscript should be published in its current form.

Reviewer 2 Report

No additional comments besides those outlined on my first review.

Reviewer 3 Report

In the study the authors describe 7 patients with CVID who have insufficient trough IgG levels despite adequate ivIG replacement therapy and underwent scintigraphy. The authors have addressed the issues raised before adequately, the overall quality of the manuscript has been improved and the clinical implications of these invasive investigations have been addressed.

However, there are minor points that should be addressed:

- “A histologic approach to the whole bowel would also be more invasive and difficult”. I do not agree with this statement. Most of the inflammation in this study has been seen in the colon (not the small intestine), which is easily accessible using routine colonoscopy. The risk of scintigraphic using labeled leukocytes seems higher. Further, it would be interesting to compare the scintigraphic finding to histology.

- The authors say that switching to SCIG improved trough IgG levels. Does this correlate with improved scintigraphic findings?